# Design and Implementation of Fast Locking All-Digital Duty Cycle Corrector Circuit with Wide Range Input Frequency

**Shao-Ku Kao** 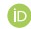

Department of Electrical Engineering and Green Technology Research center, School of Electrical and Computer Engineering, College of Engineering, Chang Gung University, Taoyuan City 330, Taiwan; kaosk@mail.cgu.edu.tw

**Abstract:** This paper presents a fast locking and wide range input frequency all-digital duty cycle corrector (ADDCC). The proposed ADDCC circuit comprises a pulse generator and a clock generator. The pulse generator is edge-triggered by an input signal to produce a 0 degree and 180 degree phase. The clock generator uses a 0 degree and 180 degree phase to produce the 50% duty cycle output signal. It corrects the duty cycle of the input signal in six clock cycles. The proposed ADDCC is implemented in a 0.35 μm CMOS process. The circuit can operate from 10 MHz to 100 MHz, and accommodates a wide range of input duty cycles ranging from 30% to 70%. The duty-cycle error of the output signal is less than ±1%.

**Keywords:** fast locking; all-digital; duty cycle corrector (DCC); wide range correction

---

## 1. Introduction

In recent years, the speed requirement for System-on-Chips (SoCs) has increased, due to the development of CMOS process technology, with clock signals widely used in most digital and mixed-signal circuits. The accuracy of the frequency, phase, and pulse width of the clock signal is important for circuit applications. However, the phase locked loop (PLL) [1–3] or delay locked loop (DLL) [4,5], can only correct the signal frequency and phase. A clock with a 50% duty cycle is very important in many applications, such as double data rate synchronous dynamic random access memory (DDR-SDRAM) and double-sampling analog-to-digital converters (ADC). To double the data rate, both positive and negative transition edges of a clock are utilized. However, the duty cycle distortion of a clock occurs owing to the unmatched rising time and falling time in the clocking paths. Thus, a duty cycle corrector (DCC) for a clock is needed [6–13].

There are two categories to implement the DCC in the literature: the feedback type [6,7] and the non-feedback one [8–14]. The design method is divided into an analog approach [6,11] and a digital approach [7–10,12–14]. One of the digital non-feedback DCCs require only a single-ended clock and achieve a fast locked time. In most lectures, the time-to-digital conversion (TDC) is used to detect the period or duty cycle of the input clock. However, the TDC is implemented with a delay line to quantize the period or duty cycle. This is a limited range of correction. To extend the range, more delay cell is needed in the delay line. The DCC of Reference [13,14] is mainly used to improve the problem of the half-cycle delay line mismatch, using a dual-loop TDC for quantization to achieve fast locking, and uses two or three half delay lines to solve the delay line mismatch problem. To further reduce the mismatch, the detection circuit is integrated with the delay line.

In this paper, an all-digital fast-locked DCC is presented to correct the duty cycle. The proposed DCC generates the 50% duty cycle within six clock cycles. The proposed circuit is implemented with a non-input delay line oscillator to achieve coarse quantization. The wide frequency range is provided by the delay line oscillator. The delay line provides a high accuracy to generate a 50% duty cycle. Therefore, the frequency of the input signal is from 10 MHz up to 100 MHz. This paper is organized as follows: Section 2 gives the circuit

description. The experimental results are given in Section 3. Finally, the conclusions are given in Section 4.

## 2. Circuit Description

In this paper, the duty cycle corrector is presented to achieve two functions: generate a 50% duty cycle and achieve wide range correction. The proposed circuit uses the delay line oscillators to transfer the duty cycle of the input signal into digital code. A new control stage is proposed to achieve a wide range correction for duty cycle correction. The proposed all-digital DCC operates with the frequency range of 10–100 MHz and a 30–70% duty cycle. The proposed circuit has been fabricated in a CMOS 0.35-μm process.

The concept of the proposed circuit is generated by the CLK (Clock)-rising and CLK-falling from the pulse generator. Then, the signal CLK-out is formed by merging the signal CLK-rising and the CLK-falling in the clock generator. The CLK-out obtains a 50% duty cycle.

The proposed circuit is composed of a control unit, a time to digital converter (TDC), a ring-active circuit, a larch & half code circuit, a digital-to-time converter (DTC), two one-shot circuits, and an SR (Set/Reset)-latch, as shown in Figure 1. The proposed ADDCC locking process has three steps: quantize the period of the input clock, select the phase, and generate the output clock. In the first step, when the input clock CLK-in passes through the control unit, it generates the start and stop signals, respectively. The pulse width of the start signal is the period of the CLK-in. The stop signal is the second rising edge of the CLK-in. The start signals input to the TDC, and produces the coarse code and the fine code, as shown in Figure 2. The stop signal latches the coarse code and the fine code. In the second step, the coarse code and the fine code shift one-bit code to obtain a Half Coarse code and a Half Fine code. The CLK-rising or the CLK-falling is selected according to the Half Coarse code and Half Fine code. The third step, by merging the CLK-rising and CLK-falling, produces a 50% duty cycle output clock CLK-out.

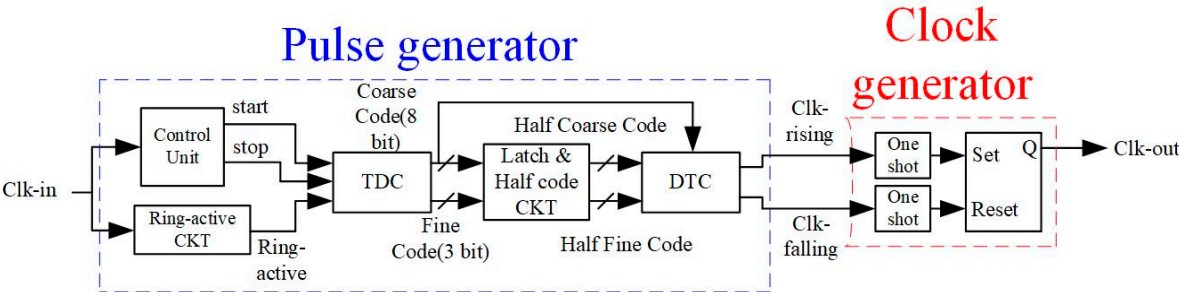

**Figure 1.** Architecture diagram of the proposed ADDCC. (CKT: Circuit).

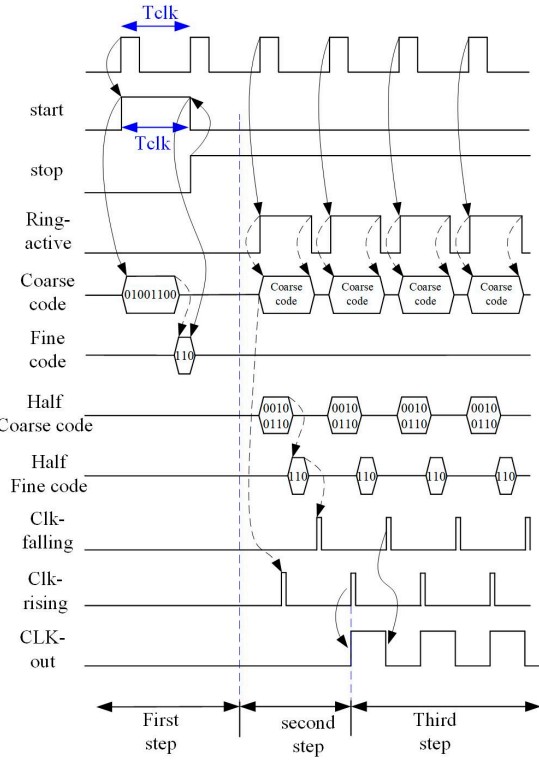

**Figure 2.** Timing diagram of the proposed ADDCC. (Tclk: Period of input clock).

## 2.1. TDC

The TDC is composed of the Fine TDC, and the Coarse TDC, as shown in Figure 3. The purpose of TDC is to quantize the period of input clock into digital code.

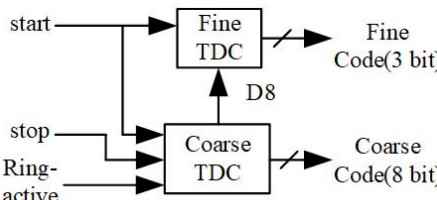

**Figure 3.** Diagram of TDC.

### 2.1.1. Fine TDC

The Fine TDC is composed of DFFs (D-type flip-flop), the SR-latch, the delay-line, the DFF array, the inverter, and the encoder, as shown in Figure 4a. The Fine TDC timing diagram is shown in Figure 4b. After the Coarse TDC, there is a phase difference between the rising edge of the D8 (Delayed-phase 8) and the falling edge of the start signal. The Fine TDC is to quantize this phase difference. The Fine signal is formed by taking the falling edge of the start signal and the first rising edge of the D8 behind the falling of the start signal. The Fine signal is quantized by the delay line and the DFF array. The multi-phases are generated by sending the Fine signal into the delay line to form Ph1 (Phase 1)–Ph8 (Phase 8). The output of the DFF array passes through the decoder and inverter (INV), and generates the find code, F [1:3]. The accurate phase difference (which is not quantized by the Coarse TDC) should be the rising edge of the D8 before the falling edge of the start signal. However, due to the rising edge of the D8, it is hard to detect before the falling edge of the start signal. Therefore, the Fine TDC quantizes the phase behind the falling edge of the start signal and the first rising edge of the D8 much more easily. To obtain the correct code, the inverter is needed at the output of the encoder.

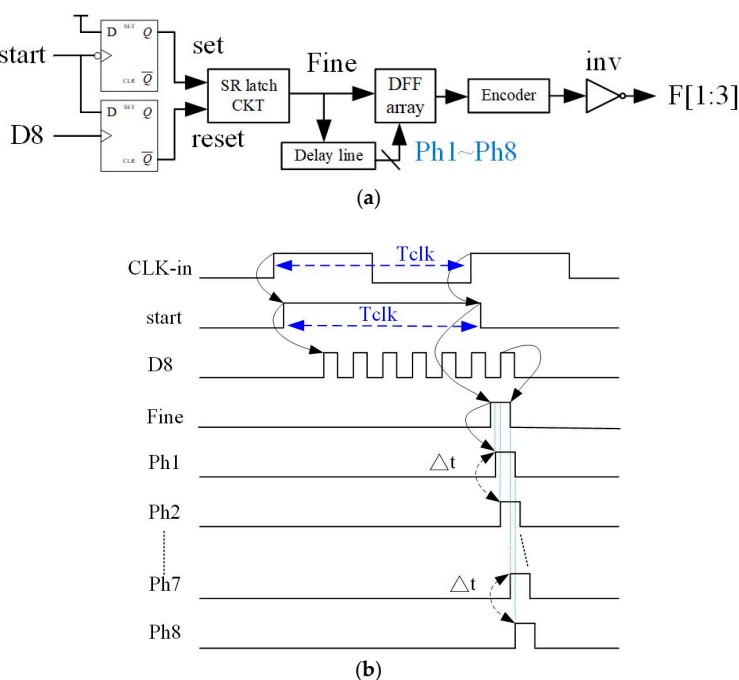

**Figure 4.** (**a**) The block diagram and (**b**) the timing diagram of the Fine TDC.

### 2.1.2. Coarse TDC

The Coarse TDC is composed of MUX (Multiplexer), delay line oscillators, and an 8-bit counter, as shown in Figure 5a. The Coarse TDC timing diagram is shown in Figure 5b. The Coarse TDC is implemented with a delay line oscillator to extend the frequency range. The period of CLK-in is quantized by the pulse signal D8, to convert into a digital code or a coarse code. The counter counts the number of the rising edge that was generated in the period of CLK-in.

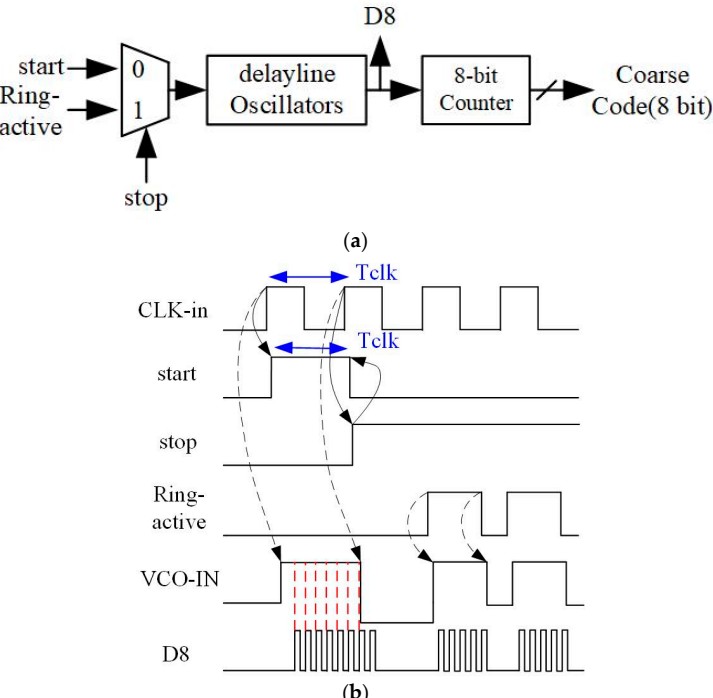

**Figure 5.** (**a**) The block diagram and (**b**) the timing diagram of the Coarse TDC. (VCO: Voltage-controlled oscillator).

### 2.2. DTC

The Divider is composed of the compare, the delay line, and the 8 to 1 MUX, as shown in Figure 6.

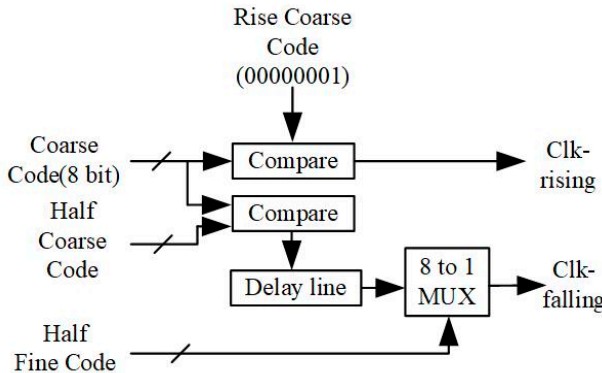

**Figure 6.** The block diagram.

### 2.3. Ring-Active CKT

The ring-active CKT is composed of DFFs, a buffer-line, and an XOR (Exclusive-OR gate), as shown in Figure 7a. When the input clock (CLK-in) passes through the Divider, it will do Divider 2 behaviors and produce D-1 and D-2 (Output of DFF) signals. After the D-1 and D-2 input delay line creates duty cycle reduction, it produces DB-1 and DB-2 (Delay pulse) signals. Finally, using XOR combined with DB-1 and DB-2 produces the periodic Divider out signal. The Divider timing diagram is shown in Figure 7b.

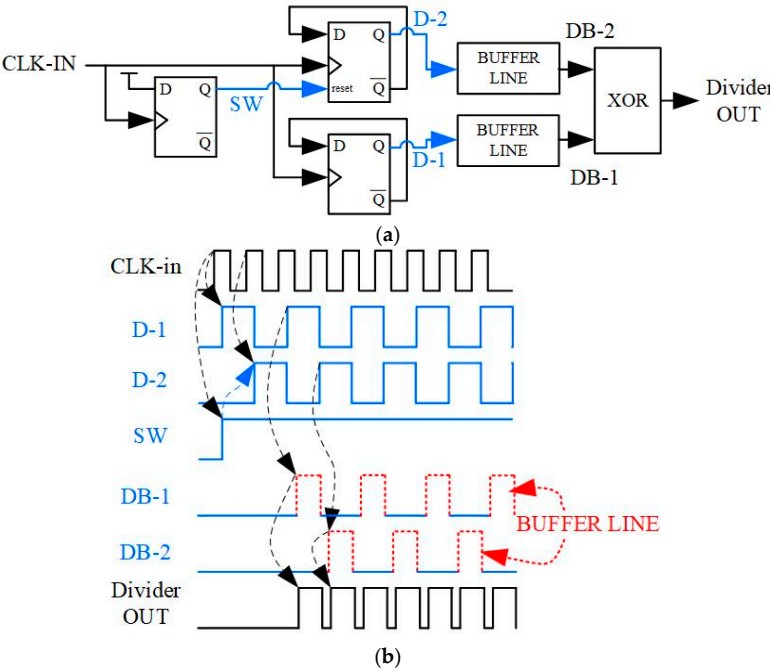

**Figure 7.** (**a**) The block diagram and (**b**) the timing diagram of the Divider.

### 2.4. Delay Line Oscillators & Delay Line

The delay line oscillator [15] is composed of a buffer, as shown in Figure 8a. It mainly has the D8 signal for the Fine TDC and the Counter CKT. The delay line oscillators are realized by the D1–D8 phase blending technique, producing the D8 oscillate signal for the Coarse TDC. The delay line oscillators timing diagram is shown in Figure 9a. Figure 8b shows the delay line circuit, which mainly provides the multi-phases (Ph1–Ph8) for the

Fine TDC; the timing diagram is shown in Figure 9b. The oscillator and delay line have the same structure as the delay cell, as shown in Figure 8a. The proposed delay line oscillator with two inverters makes the frequency of oscillation the same as the phase difference between the input of the delay line and Ph8. The delay line oscillator circuit and delay line circuit can generate the same frequency. Besides this, both circuits have the same effects on PVT (process, voltage temperature) variations.

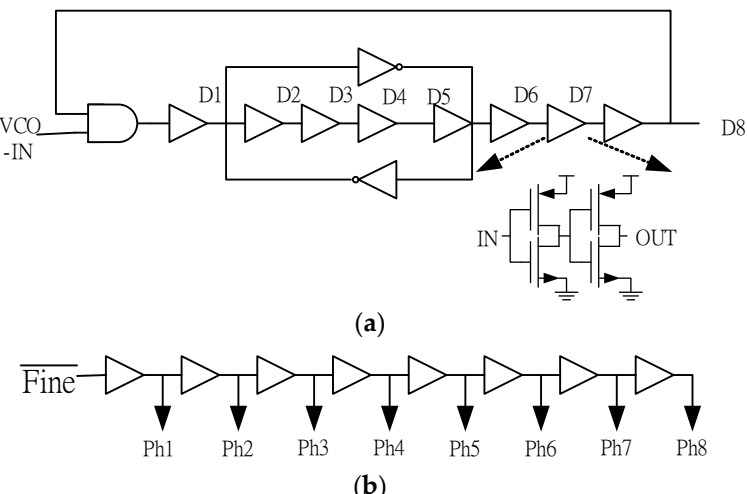

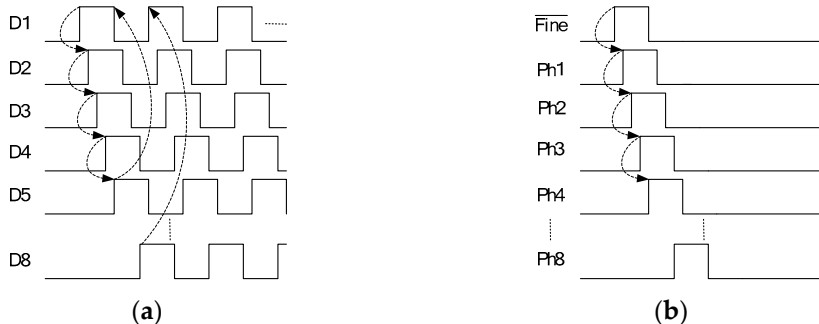

**Figure 8.** (**a**) Delay line oscillator circuit and (**b**) delay line circuit.

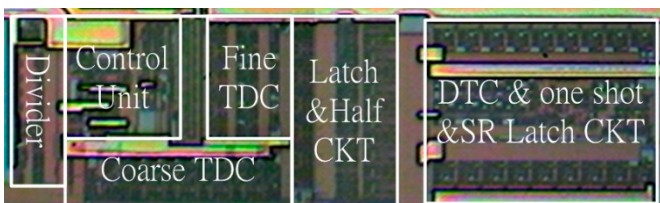

**Figure 9.** (**a**) Delay line oscillators and (**b**) delay line of timing diagram.

## 3. Experimental Results

The proposed all-digital 50% DCC was fabricated in the 0.35 μm CMOS process. The circuit was tested from a 3.3 V supply voltage. Figure 10 shows the all-digital 50% DCC chip photo. This fully-integrated chip, including pads, occupies an area of 0.28 × 0.96 mm.

**Figure 10.** Chip photo.

Due to the open-loop architecture of the DCC, the DCC operation is completed within six clock cycles, as illustrated in Figure 11, which is the measurement waveform for the clock frequency of 100 MHz and a ±1% output duty error.

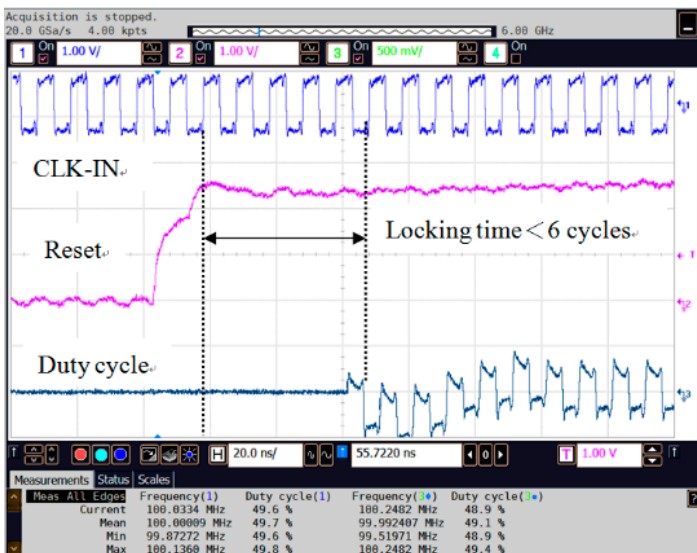

**Figure 11.** Locking time measurement results.

The acceptable duty cycle of the input clock ranges from 30 to 70%, and the clock frequency from 10 MHz to 100 MHz. The waveform of the duty-cycle corrected output clock is shown in Figure 12a, where the input clock frequency and the duty cycle are 100 MHz and 30%, respectively. Figure 12b shows the input clock frequency and duty cycle at 100 MHz and 70%, respectively. The jitter histogram in Figure 13 exhibits 5.4 ps RMS (Root Mean Square) jitter and 49.4 ps peak-to-peak jitter. Figure 14 also gives the measured duty-cycle errors with respect to different input duty cycles at 10 MHz–100 MHz. The P-P (Peak to Peak) jitter and the RMS jitter histogram are shown in Figure 15. The comparisons among the proposed circuit and several previous works are listed in Table 1. The clock's duty cycle plays an important role in systems such as DDR-SDRAM, double-sampling ADC, and clock and data recovery, which requires the rising and falling edges of a clock. Therefore, the proposed DCC circuits can be implemented in the system to keep a 50% duty cycle.

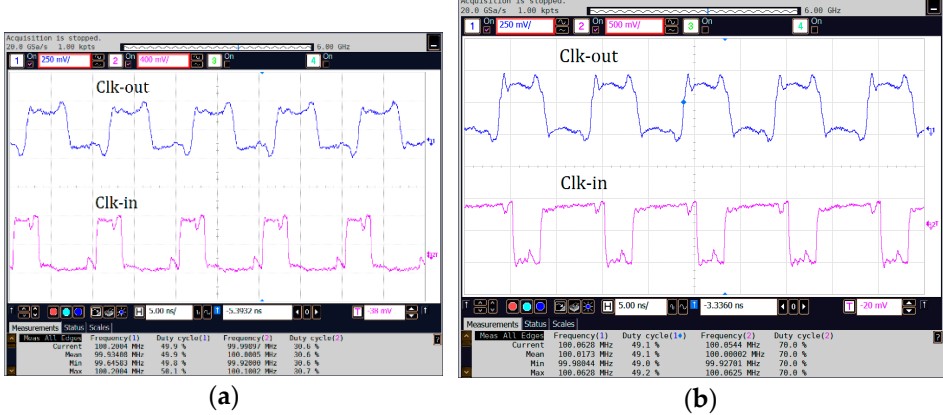

**Figure 12.** Measurement waveform of output clock for clock frequency of 100 MHz (**a**) input duty cycle of 30% (**b**) input duty cycle of 70%.

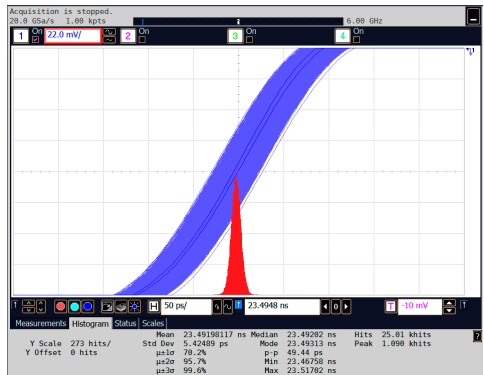

**Figure 13.** Output jitter histogram for a 100 MHz signal.

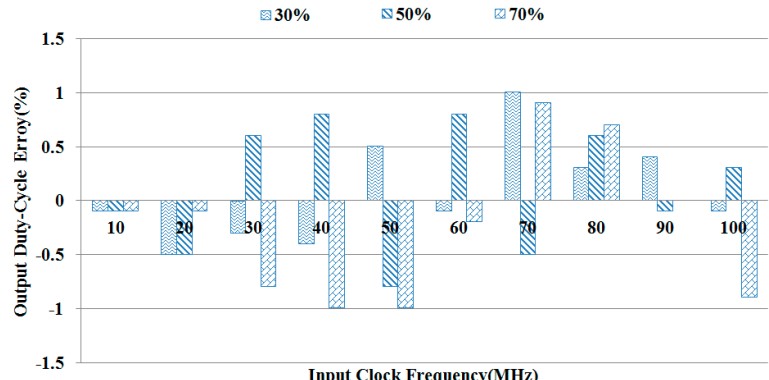

**Figure 14.** Measured duty-cycle errors for different input duty cycles at 10 MHz–100 MHz.

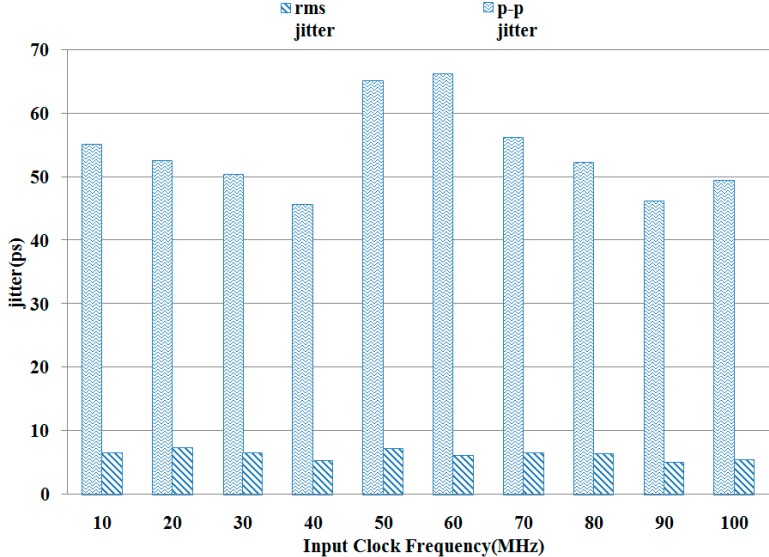

**Figure 15.** Measured P-P jitter and RMS jitter for different input duty cycles at 10–100 MHz.

**Table 1.** Comparison with previous works.

|  | [10] | [11] | [12] | **This Work** |
|---|---|---|---|---|
| Type | Digital | Digital | Digital | Digital |
| Process (μm) | 0.35 | 0.25 | 0.35 | 0.35 |
| Supply Voltage (V) | 2.5 V | NA | 3.3 V | 3.3 V |
| Frequency range (MHz) | 250–600 | 400 | 400–600 | 10–100 |
| Duty cycle range | 40–60% | 2–98% | 30–70% | 30–70% |
| Total locked time | <5 cycle | <30 cycle | <28 cycle | ≤6 cycle |
| Duty cycle error | ±0.8% | −1.2–2.2% | ±0.6% | ±1% |
| Jitter (ps) | 64.4 ps @600 MHz | NA | 16.7 ps @500 MHz | 49.44 ps @100 MHz |
| Core size (mm$^2$) | NA | NA | 0.68 | 0.27 |

Table 1 shows a comparison with previous studies. The proposed circuit has a wider input frequency range and is fast locking. The highest operating frequency of this circuit is mainly limited by the delay time of the delay line. The lowest operating frequency is limited by the bits of the counter inside the Coarse TDC. Although increasing the delay line length can allow the circuit to operate at lower frequencies, this substantially increases the power consumption of the chip.

## 4. Conclusions

An all-digital fast locking DCC is presented. The duty cycle is corrected in six cycles. The proposed all-digital DCC operates with a frequency range of 10–100 MHz and a 30–70% duty cycle. The highest operating frequency of this circuit is mainly limited by the delay time of the delay line. The delay time of the delay cell can reduce with advanced technology. The measured peak-to-peak and root mean square are 49.4 ps and 5.4 ps, respectively, at 100 MHz under a 3.3 V. The measured duty-cycle error is between 1% and −1% with 30–70% input duty cycles at 10–100 MHz. The proposed circuit has been fabricated in a CMOS 0.35-μm process.

**Author Contributions:** Conceptualization, writing—original draft preparation, writing—review and editing are done by S.-K.K. The author has read and agreed to the published version of the manuscript.

**Funding:** This research received no external funding.

**Institutional Review Board Statement:** Not applicable.

**Informed Consent Statement:** Not applicable.

**Data Availability Statement:** Data sharing is not applicable to this article.

**Acknowledgments:** This work was supported in part by the Ministry of Science and Technology (MOST), Taiwan, under Grant MOST 109-2221-E-182-053. The authors would like to thank the EDA tool supports and the chip manufacturer of Taiwan Semiconductor Research Institute (TSRI).

**Conflicts of Interest:** The author declares no conflict of interest.

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
