# Peer review of "Design and Implementation of Fast Locking All-Digital Duty Cycle Corrector Circuit with Wide Range Input Frequency"

_electronics, doi:10.3390/electronics10010071_

Round 1
Reviewer 1 Report
The paper presents a fast locking and wide range input frequency all-digital duty cycle corrector. The form of this paper in unacceptable. It is just short engineering letter. You show the implementation but what is the scientific value? It is very poor. This paper should be rewritten from scratch.
1. The introduction is too short. The state of the art is rather poorly presented.
2. Your design should be compared with other known methods. The scientific value (novelty) should be exposed is the section 2. Pros and cons of other methods should be presented and juxtaposed with your proposals. The main contribution should be presented here.
3. Experimental results are commented briefly. The table 1 is almost without comment. Why do you think someone should applied your DCC? In what application will your DCC be better than others?
4. Conclusions should highlight the value of your design in relation to other DCCs. Pros and cons with novelty should be presented.
However, the implementation of the designed circuit deserves praise.
Author Response
Dear reviewer:
Thank you very much for giving us an opportunity to revise our manuscript. We appreciate the editor and reviewers very much for their constructive comments and suggestions on our manuscript entitled “ Design and Implementation of Fast Locking all-digital duty cycle corrector circuit with wide range input frequency”
We have studied reviewers’ comments carefully. Those comments are very helpful for revising and improving our paper, as well as the important guiding significance to other research. According to the reviewers’ detailed suggestions, we have made a careful revision on the original manuscript. All revised portions are marked in red in the revised manuscript and the responds to the reviewers’ comments are as follows.

Reviewer 2 Report
- Abstract is clear and precise.
- Introduction section provides brief but useful background about the work.
- 1 shows block diagram of the overall architecture of ADCC that makes sense.
- Caption used for figure 1 is “fig. 1”, whereas, all other captions used are “figure” . Please be consistent with figure captions.
- Overall, this work is interesting and useful. The text requires some proof reading though.
Author Response
Dear reviewers:
Thank you very much for giving us an opportunity to revise our manuscript. We appreciate the editor and reviewers very much for their constructive comments and suggestions on our manuscript entitled “ Design and Implementation of Fast Locking all-digital duty cycle corrector circuit with wide range input frequency”
We have studied reviewers’ comments carefully. Those comments are very helpful for revising and improving our paper, as well as the important guiding significance to other research. According to the reviewers’ detailed suggestions, we have made a careful revision on the original manuscript. All revised portions are marked in red in the revised manuscript and the responds to the reviewers’ comments are as follows.

Reviewer 3 Report
The article discusses the Design and Implementation of Fast Locking all digital duty cycle corrector circuit with wide range input frequency. The introduction, Experimental, and conclusion part need more scientific information. The manuscript needs more organization with relevant information with reasonable evidence. However, the following comments must be addressed prior to the publication.
Comments:
- Ln No 21: write the abirritation of DDR-SDRAMs. The first time in the introduction part or experimental part should write the full abbreviation and then authors may use the short form.
- The introduction part is very small with little information. The author must elaborate and write more applications along with current or very recent references. Adding more reference is must and qualify the current work.
- Ln No :114: Fig 8. Delay line oscillators circuit [6]- Delayline is one ward or Delay -line. Also, remove the [6]
- Ln no 118: Authors may represent the real experimental picture
- Ln no 128: Author representing Figure 11. Locking time Measurement results. Include the results in the number in the table. Same thing for figure 13 as well.
- Ln no 151: Conclusion: Should rewrite the current results in more detail along with application and how the present work is going to improve the current research.
Author Response

(The authors gave the same response as above.)

Round 2
Reviewer 1 Report
I accept the corrections, but, in my opinion, the scientific value of the article is still low. In the future, please show the novelty in relation to the existing methods.
Author Response

(The authors gave the same response as above.)
